# Recurrent Glioblastoma: A Review of the Treatment Options

**DOI:** 10.3390/cancers15174279

**Published:** 2023-08-26

**Authors:** Maria Angeles Vaz-Salgado, María Villamayor, Víctor Albarrán, Víctor Alía, Pilar Sotoca, Jesús Chamorro, Diana Rosero, Ana M. Barrill, Mercedes Martín, Eva Fernandez, José Antonio Gutierrez, Luis Mariano Rojas-Medina, Luis Ley

**Affiliations:** 1Medical Oncology Department, Ramon y Cajal University Hospital, 28034 Madrid, Spain; mvilladelga88@gmail.com (M.V.); vicalbarranfernandez@gmail.com (V.A.); victor.alia@salud.madrid.org (V.A.); pilar.sotoca@salud.madrid.org (P.S.); jchamorro@salud.madrid.org (J.C.); di.rosero328@uniandes.edu.co (D.R.); anamaria.barrill@gmail.com (A.M.B.); 2Radiotherapy Oncology Department, Ramon y Cajal University Hospital, 28034 Madrid, Spain; mercedes.martin.sanchez@salud.madrid.org (M.M.); efernandezl@salud.madrid.org (E.F.); 3Neurosurgery Department, Ramon y Cajal University Hospital, 28034 Madrid, Spain; joseantugui1@hotmail.com (J.A.G.); luismarianorojas@gmail.com (L.M.R.-M.); luis.ley@salud.madrid.org (L.L.)

**Keywords:** glioblastoma, recurrent glioblastoma, brain tumor, glioblastoma chemotherapy, immunotherapy, glioblastoma radiotherapy, glioblastoma surg

## Abstract

**Simple Summary:**

Glioblastoma is the most common malignant brain tumor associated with a poor prognosis, with a median survival of 14 months. Despite initial treatment with surgery, radiotherapy, and chemotherapy, recurrence is the usual situation. Controversy remains over the best treatment strategy for recurrent disease, and there is no standard of treatment in this situation. Different forms of treatment have been addressed, including a surgical procedure or radiotherapy, systemic treatment with chemotherapy or targeted drugs, and different immunotherapy strategies. Knowledge of the data from these studies allows for improved decision-making in this clinical situation.

**Abstract:**

Glioblastoma is a disease with a poor prognosis. Multiple efforts have been made to improve the long-term outcome, but the 5-year survival rate is still 5–10%. Recurrence of the disease is the usual way of progression. In this situation, there is no standard treatment. Different treatment options can be considered. Among them would be reoperation or reirradiation. There are different studies that have assessed the impact on survival and the selection of patients who may benefit most from these strategies. Chemotherapy treatments have also been considered in several studies, mainly with alkylating agents, with data mostly from phase II studies. On the other hand, multiple studies have been carried out with target-directed treatments. Bevacizumab, a monoclonal antibody with anti-angiogenic activity, has demonstrated activity in several studies, and the FDA has approved it for this indication. Several other TKI drugs have been evaluated in this setting, but no clear benefit has been demonstrated. Immunotherapy treatments have been shown to be effective in other types of tumors, and several studies have evaluated their efficacy in this disease, both immune checkpoint inhibitors, oncolytic viruses, and vaccines. This paper reviews data from different studies that have evaluated the efficacy of different forms of relapsed glioblastoma.

## 1. Introduction

Glioblastoma (GB) is the most common primary brain tumor. The incidence is 3.2 cases per 100,000 population, with a peak in older than 65 years old. The median overall survival (OS) since diagnosis in treated patients is about 14 months [1].

As outlined, this is one of the most aggressive tumors, and recurrence is almost unavoidable. The mean progression-free survival (PFS) is approximately 7 months since diagnosis [2].

Recurrent GBs are associated with a poor prognosis (median OS less than 1 year), and there is no standard of care in this clinical situation.

In this situation, the treatments are considered not curative, and few randomized trials have addressed the question of the best treatment option. The majority are retrospective or non-randomized trials, and therefore, the decision of the best treatment option should be individualized and take into account the risks and benefits, as well as the quality of life aspects, in the context of non-curative disease.

Different approaches should be considered, including loco-regional treatments such as surgery, radiotherapy (RT), or devices like Novo TTF-100A.

Also, systemic agents have been studied in the case of recurrent disease. Chemotherapy (CT), particularly nitrosoureas, antiangiogenic drugs (bevacizumab), tyrosine kinase inhibitors (TKI), and immunotherapy, have been the main focus of the developed research.

Here, we will review the most recent scientific evidence for the treatment of relapsed GB.

## 2. Surgery

The rate of patient candidates for a new surgery is approximately 20–30% [3]. The median OS observed is approximately 9 months [4].

It has been reported that patients who are re-operated are usually younger, with better Karnofsky performance status (KPS ≥ 70) and smaller tumor size [5].

The main prognostic factors after a second surgery are the performance status (KPS), age, and extent of the surgical resection (EOR) [4]. Time to progression is one of the most frequent factors considered to decide on a new surgery. Those with early recurrence used to have a greater risk of death than patients with prolonged PFS [6]. Indeed, the median time from the first to the second surgery is approximately 1 year [7].

Nevertheless, there is no clear consensus on selecting which case will benefit the most from a re-intervention. The key to decision-making is to improve quality of life and also overall survival. Park et al. developed a three-tier scale composed of scores for KPS and ependymal involvement to decide the patients that will benefit most from the surgical resection [8].

Another developed scale included tumor involvement of prespecified eloquent/critical brain regions, KPS (≤80), and tumor volume (≥50) [9].

There are limited studies on the impact of surgery on survival (OS) in recurrent glioblastoma. Robin and colleagues [4] 2017 published a comprehensive literature review that included 33 studies evaluating the benefit of surgery on OS at progression after a first line using the Stupp protocol. Most of them showed a positive impact on the OS of re-operation, with an estimated median OS of 9.9 months. Nevertheless, most studies were retrospective (only six were conducted prospectively), and there were no randomized controlled trials. In 10 of 33 evaluated studies, no survival benefit was observed.

In fact, in contrast to previous studies where a survival benefit has been observed, Sastry et al. reviewed a cohort of 368 patients, founding, after multivariate analysis, that surgery may not improve survival after tumor progression in the context of contemporary non-surgical therapy. Three variables were identified to prolong OS, KPS > 80, bevacizumab, and cytotoxic chemotherapy at first progression but not surgery for progression, suggesting that when controlling potentially confounding factors as well as the introduction of more recent treatments in the analysis, the benefit of resection in recurrent GB is not statistically significant [10]. Nevertheless, the authors concluded that beyond the effect on survival, the benefit of surgery in recurrence is based on reducing neurological symptoms or obtaining a new histological simple.

Several reports have questioned if these results could be biased by the above-mentioned impact factors (age, KPS, or lesion characteristics).

The EOR at recurrence is another important issue. A gross total resection (GTR) at first surgery has clear positive outcomes in PFS, OS, quality of life, and symptomatic control [11]. At reoperation, Lu and colleagues [12] observed that maximal resection was significantly associated with longer OS (HR, 0.59; 95% confidence interval [CI], 0.43–0.79; heterogeneity < 0.01). Other studies support better results in OS after GTR [13].

Even so, achieving a GTR is more difficult when there is a progression, with more post-surgical complications (18.9%) or neurological sequelae [4]. For this reason, the aim is to perform surgery as extensively as possible while preserving neurological function and good quality of life [7]. Different techniques have been developed, such as fluorescence-guided surgery, using 5-aminolevulinic acid (5-ALA), where tumor cells are able to emit fluorescence recognized by some filters facilitating a complete resection [14] or awake craniotomy to remove, with higher security, contiguous lesions to cortical or subcortical eloquent areas.

In the literature review done by Montemurro et al., there are some studies that found the benefit of surgery followed by systemic treatment, with an increase in OS from 6 to 14 months with the use of adjuvant CT in the study of Bonis et al. [15]. Previous publications reported the benefit of adding carmustine wafer implants on resected cavities [6].

On the other hand, complications of the surgical procedure should also be considered in the decision-making process, estimated at up to 10% mortality and a range of 13–69% morbidity [4].

The ongoing RESURGE study is focused on the role of surgery in the treatment of relapsed GB, comparing tumor resection followed by adjuvant second-line therapy to no surgery. OS, neurological status, and quality of life will be analyzed.

Therefore, surgery is one of the most valuable tools in treating recurrent glioblastoma, allowing symptom control in addition to its effect on survival control. Patient or tumor characteristics (youth and good performance status or accessible lesion) may be the most important impact factors in overall survival. Even so, more studies are needed to confirm the role of surgery in these patients.

## 3. Radiotherapy (RT)

In general, RT treatment is rarely considered at the time of GB relapse. One of the most important aspects is that the majority of relapses occur within the high-dose radiation field (90–95%), so toxicity concerns should be taken into consideration.

A proper selection of patients to be treated will be a key issue. The factors most frequently considered to indicate treatment with radiotherapy are KPS, age, time to progression, type of progression, target volume, and site of recurrence.

Therefore, the role of reirradiation is uncertain.

In addition, there are few prospective data. A recent prospective phase II study included 90 patients for re-irradiation in high-grade gliomas. The median OS was 17 months, radionecrosis occurred in 10% of cases, and neurocognitive functions remained stable. The authors concluded that re-irradiation can be considered a feasible option with low toxicity. However, the inclusion of other histological types, such as oligodendrogliomas and grade 3 astrocytomas, may also have conditioned these results [16]

A meta-analysis on retrospective studies [17] included 2095 patients from 50 non-comparative studies treated with re-irradiation after glioblastoma recurrence. The results were OS-6 months 73%, OS-12 months 36%, OS-PFS-6 months 43%, and PFS-12 months 17%. Higher OS-12 months was observed in studies using brachytherapy, and a tendency to better PFS-6 months with moderate hypofractionation schedules (≤5 fractions). Within the external beam radiotherapy group, no differences were found between equivalent doses at 2 Gy per fraction (EQD2 doses) > or < 36 Gy. In terms of adverse effects, grade 3 toxicity occurred in 7% of patients (not collected in the brachytherapy studies). It can, therefore, be concluded that it is a feasible treatment option after appropriate patient selection.

Regarding patient selection, Straube et al. [18] established which patients were candidates for re-irradiation: non-candidates for surgery (tumors in eloquent, unresectable areas), KPS > 50, and > 6 months since the first irradiation.

Scoccianti [19] lists as factors to consider prior to re-irradiation: The recursive partitioning analysis (RPA) class, monofocality, target volume, time since first irradiation (best >12 months), tumor grade, and age (<50 years). Knisely et al. [20] considered factors in favor of re-irradiation: young patients, good performance status, >12 months since the first irradiation, recurrence outside the previous irradiation field, small and localized, and in areas of lower sensitivity to radiation.

The doses, techniques, and volumes of treatment are not clearly established. Scoccianti [19] proposes doses and radiotherapy techniques depending on the volume to be irradiated: in cases of <12.5 cc, radiosurgery at a dose of 12–15 Gy; 12.5–35 cc, hypofractionated radiotherapy 25 Gy in 5 fractions of 5 Gy and in cases of 35–50 cc, conventional fractionation at 36 Gy in 20 fractions. They recommend contouring to delimit the volume of the lesion in T1 sequences with contrast and give margins ≤5 mm to generate the planning target volume (PTV).

Minniti [21] also establishes a dose and technique depending on the volume to be treated: 4–10 cc, radiosurgery with a dose of 15–18 Gy; 8.5–34 cc, moderate hypofractionation (35 Gy in 10 fractions, e.g.); 33–145 cc, extreme hypofractionation (25 Gy in 5 fractions, e.g.); and if >100 cc conventional fractionation (36 Gy in 2 Gy/fraction). Regarding the volume of treatment margins for general PTV, they recommend 1 cm for conventional fractionation, 5 mm for hypofractionation, and no margin for radiosurgery.

Another frequently used scheme is 30 Gy in five fractions. This fractionation has been explored in a recent meta-analysis conducted by Luo et al. [22], analyzing the results of 301 patients using this fractionated stereotactic scheme. They reported a 12-month OS of 33.1% and a 12-month PFS of 13.4%. Age less than 55 years old, time to recurrence longer than 11. 2 months, total dose less than 30 Gy, and single dose more than 5 Gy are influencing factors for better OS and PFS.

Radiotherapy with protons has also been studied in the reirradiation scenario in an attempt to better preserve cerebral parenchyma from radionecrosis. In the absence of comparative data with photon therapy, the largest series is reported in a prospective multicentric study [23] with 45 patients with recurrent glioblastoma re-irradiated with proton therapy. The median PFS of 13.9 months, and the median OS of 14.2 months. One patient had acute grade 3 toxicity, and four patients had late grade 3 toxicity. In another study evaluating the quality of life in the recurrent glioblastoma setting with protons, Scartoni et al. [24] reported data from 33 patients showing that quality of life parameters are stable during and after treatment until the time of new progression, so they conclude that proton therapy is safe and well tolerated.

A survival benefit has been suggested as a benefit in several observational studies using interstitial brachytherapy in patients with recurrent high-grade gliomas. However, an association with a high incidence of radiation necrosis has been found [25,26].

An alternate form of brachytherapy uses an inflatable balloon catheter containing a liquid I-125 radioisotope (GliaSite) inserted at the time of surgical resection, allowing the delivery of a high dose of radiation to the tissue. However, no comparative studies with other techniques have been carried out [27]. In addition, the role of brachytherapy is diminishing as experience with stereotactic radiosurgery (SRS) and fractionated localized limited field radiation evolves.

Reirradiation can be given both with concurrent or sequential administration of systemic therapy (TMZ, bevacizumab, immunotherapy, and others). Only a few prospective studies are available.

In a retrospective study, Baehr et al. evaluated in 46 patients the combination of reirradiation with systemic therapy (TMZ, bevacizumab, nitrosoureas, and others). TMZ improved PFS (6.6 vs. 4 months *p* < 0.001) and OS (17 vs. 10 months *p* = 0.1) vs. all other systemic therapies [28].

In a phase II trial, 182 patients with recurrent glioblastoma were randomized to receive bevacizumab alone or in combination with radiation treatment. A benefit was found for the combined treatment in PFS (7.1 vs. 3.8 months *p* = 0.05), without significant improvement in OS [29]. A smaller single-center trial compared bevacizumab-based chemotherapy with or without fractionated radiosurgery (32 Gy in four fractions) in 35 patients with recurrent high-grade glioma. Reirradiation had improved progression-free survival (5.1 versus 1.8 months) and no benefit in OS (7.2 versus 4.8 months, *p* = 0.11). Adverse effects were similar between groups, and there were no cases of radiation necrosis [30].

A recent publication of a systematic review showed data on 1399 patients treated with RT alone or RT and bevacizumab. The combined treatment was associated with significantly improved OS (2.51 vs. 4.92 months), *p* = 0.04, but no significant improvement in PFS (1.40 vs. 3.18 months), *p* = 0.099. In addition, the combination had significantly lower rates of radionecrosis (2.2% vs. 6.5%, *p* < 0.001) [31]. Other initial trials (phase I) studied the combination of RT, bevacizumab, and immunotherapy with promising results, but further controlled studies are needed to confirm these effects [32].

Whether or not there is an improvement with the addition of adjuvant radiotherapy treatment after resection of glioblastoma recurrences is also not established.

In a retrospective study, 84 patients were analyzed with recurrent high-grade gliomas who underwent reoperation. Forty-two of these patients were subsequently treated with reirradiation. In patients with two or three risk factors (age > 50, WHO grade IV, unmethylated MGMT promoter), OS and PFS were significantly better after both treatments compared with surgery alone [33]. Straube et al. comment on the possible efficacy of early re-irradiation after gross total resection with the aim of decreasing the multitude of relapses after complete resection [34].

There is currently an ongoing GLIOCAVE trial [35] that will try to answer this question.

## 4. Systemic Treatment

Regarding systemic therapy, several options could be considered in the treatment of recurrent glioblastoma; all of them are considered palliative, and none have been proven to be superior to another. Therefore, the choice among these therapies will be individualized based on tumor and patient characteristics.

### 4.1. Nitrosureas

Lomustine and fotemustine are alkylating agents of the nitrosourea family. Lomustine remains one of the most widely used treatments in recurrent GM, has been the comparative arm in different clinical trials, and has the advantage of oral administration. The main frequent adverse event is thrombocytopenia.

Fotemustine is a third-generation chloroethyl nitrosourea containing a phosphoalanine carrier group grafted to the nitrosourea radical, which is able to cross the blood–brain barrier (BBB), and is even more lipophilic than lomustine [36]. The administration of this drug is intravenous, and the main side effects are also thrombocytopenia and also neutropenia.

#### 4.1.1. Lomustine

Lomustine, also known as CCNU, is an alkylating chemotherapeutic agent of the nitrosourea family that alkylates DNA and RNA and causes crosslinking of DNA and RNA, affecting their functioning, which triggers cell death in cancer cells. Since it is a lipid-soluble drug, it permeates the blood–brain barrier [37].

The study made by Yamamuro et al. proved lomustine activity in glioblastoma models, primary non-treated models, and also resistance models were included. It showed a suppressed proliferation in a dose-dependent manner, induced apoptosis, and exhibited an efficient cell-killing effect [38].

Hochberg et al. performed a retrospective study with lomustine for recurrent high-grade gliomas with a median OS of 11.5 months. The FDA approved lomustine back in 1976 for this indication [39].

It is administered orally at a dose of 80–110 mg/m^2^ once every 6 weeks [39]. In the latest European Association of Neuro-Oncology (EANO) guidelines, lomustine is considered, as well as other alkylating agents, as an option in the recurrence or progression scenario. Although there is no standardized treatment for patients with progressive glioblastoma, lomustine is being widely used in this setting, and it has also been used as a control arm in many clinical trials.

Indeed, there are many studies that used lomustine as a control arm in recurrent glioblastoma. They are compiled in the review made by Weller et al., concluding that there is a low objective response rate to lomustine in the range of 10%, a median PFS that does not exceed 2 months, a 6-month PFS of around 20%, and a median OS of 7–8.6 months. In these randomized clinical trials, lomustine is compared with many other drugs rather than bevacizumab, like enzasturin, cediranib, axitinib, galunisertib, and regorafenib. None of these experimental arms was superior to lomustine [40].

There have also been studies comparing lomustine and bevacizumab, as discussed below.

The REGOMA study is a randomized, multicentre, open-label phase 2 trial comparing regorafenib vs. lomustine. A total of 119 patients were included. OS, the primary endpoint, was significantly improved in the regorafenib group with a median OS of 7·4 months (95% CI 5.8–12.0) vs. 5.6 months (4.7–7.3) in the lomustine group (hazard ratio 0·50, 95% CI 0.33–0.75; log-rank *p* = 0.0009) [41]. Some limitations of the trial were that patients receiving regorafenib had some more favorable prognostic factors, like younger patients, more had MGMT promoter-methylated tumors, and fewer used steroids at baseline. The adverse effects rate was higher in the regorafenib arm; grade 3–4 adverse events occurred in 56% of patients treated with regorafenib and in 40% of patients treated with lomustine.

Lomustine may also be used in combination, such as in the PCV regimen, mainly used in lower-grade gliomas. The PCV regimen includes lomustine given at 110 mg/m^2^ on day 1, procarbazine 60 mg/m^2^ on days 8–21, and vincristine 1.4 mg/m^2^ on days 8 and 29 of a six-week cycle.

Lomustine is considered a safe and well-tolerated treatment in monotherapy. The emetogenic effects are well-controlled with standard prophylaxis. The most frequently reported side effect is thrombocytopenia, which is a common cause of dose reduction, treatment delays, and even suspension. Other hematologic toxicities like neutropenia are less frequent. As a non-hematologic adverse effect, alteration in liver enzymes and respiratory toxicity are reported [40].

#### 4.1.2. Fotemustine

Fotemustine is an alkylating chemotherapeutic agent, able to cross the blood–brain barrier [36].

A common dose scheme consists of fotemustine 100 mg/m^2^ e.v. weekly for 3 consecutive weeks (induction therapy), followed after 5 weeks of rest by one infusion of 100 mg/m^2^ every 3 weeks (maintenance therapy).

The Gruppo Italiano Cooperativo di Neuro-Oncologia (GICNO) conducted a phase II study for patients with recurrent or progressive glioblastoma after radiotherapy and temozolomide treatment. Forty-three patients were enrolled. PFS at 6 months (PFS-6) was 20.9% (95% CI: 9–33%), and the median OS was 6 months (95% CI: 5–7). Results by MGMT status were also reported: disease control was 75% versus 34.6% in methylated versus unmethylated MGMT patients (*p* = 0.044) [42]. The study was amended after the first three patients by decreasing the fotemustine induction dose from 100 mg/m^2^ to 75 mg/m^2^.

In the phase II, multicentre, trial by Fabrini et al., 50 patients with progressive glioblastoma after radiotherapy plus concomitant and/or adjuvant temozolomide received the previously described fotemustine regimen. PFS was 6.1 months, PFS-6 months was 52%, and median OS from primary diagnosis was 24.5 months. The efficacy control of the disease was 62% [43].

In another study by Sococcianti et al., twenty-seven patients of a single institution received fotemustine as a second-line therapy. Eight partial responses (29.6%) and five cases of stable disease (18.5%) were observed. PFS at 6 months was 48.15%. Median OS from diagnosis of glioblastoma was 21.2 months [44].

Fabi et al. conducted a retrospective study to assess the efficacy of fotemustine as a second-line treatment for recurrent glioblastoma, with the particularity that patients were classified into three groups according to the dose of fotemustine received, from 65 mg/m^2^ to 100 mg/m^2^. Groups A and B, with the lowest administered dose, showed a response rate of 40% and 26.5%, respectively, whereas patients in group C (highest dose) responded in 10% of cases. Grade 3 and 4 hematologic toxicity was only reported in Group C, thrombocytopenia being the most frequent (20%), followed by neutropenia (15%). Results by MGMT status were also reported, with a higher OS for the methylated patients (45 months) compared with unmethylated (22 months), although the MGMT status was only performed in 19 patients. The authors concluded that a low-dose fotemustine at 65–75 mg/m^2^ (induction phase) followed by 75–85 mg/m^2^ (maintenance phase) has an activity comparable to that of the conventional regimen [45].

A study published by Addeo et al. included 40 patients with recurrent glioblastoma treated with fotemustine 80 mg/m^2^ every 2 weeks for five consecutive administrations (induction phase) and then every 3 weeks at 100 mg/m^2^ as maintenance. PFS at 6 months was 61%, and the median PFS was 6.7 months (95% CI: 3.9–9.1 months). The median OS from the beginning of chemotherapy was 11.1 months. These results demonstrate that this is a better-tolerated schedule with similar efficacy data. Therefore, in many cases, this is the dose considered for patients treated with fotemustine [46].

AVAREG is a phase II, randomized, open-label, non-comparative study in which ninety-one patients with recurrent glioblastoma received bevacizumab or fotemustine (75 mg/m^2^ on days 1,8,15, then 100 mg/m^2^ every 3 weeks). The 6 months OS was 62.1% for bevacizumab (95% CI, 48.4–74.5) and 73.3% for fotemustine (95% CI, 54.1–87.7), respectively. The OS rates at 9 months were 37.9 and 46.7%, and the median OS was 7.3 months for bevacizumab and 8.7 months for fotemustine [47]. In patients with MGMT promotor methylated tumors, de OS-6 months rates were higher with fotemustine than bevacizumab.

Some trials have assessed the role of fotemustine in combination with other agents. Silvani et al. assessed the safety and efficacy of the combination of fotemustine with procarbazine. Fifty-four patients with recurrent glioblastoma were included. The median PFS was 19.3 weeks (95% CI, 14.1–24.4 weeks). The median OS from the first diagnosis was 20.8 months (95% CI, 16.7–24.8) [48]. Although these results are promising, more phase III studies are needed.

Either in monotherapy or in combined regimens, the main toxicity reported of fotemustine is thrombocytopenia, followed by neutropenia. In the study by GICNO, with the standard dose of 100 m/m^2^ at induction grades 3 and 4, thrombocytopenia was 20.9%; after the dose adjustment to 75 mg/m^2^, it was reduced to 15%. Neutropenia 3 and 4 with the adjusted dose was 15%. Other than grades 3 and 4, hematological toxicities, nausea and vomiting (4.6%), transaminase elevation (9.3%), and pneumonia were reported (2.3%).

### 4.2. Bevacizumab

Bevacizumab is a recombinant humanized monoclonal antibody that binds to the circulating vascular endothelial growth factor (VEGF). It is composed of human immunoglobulin (IG) G1 constant and murine VEGF-binding regions. The main benefit of this drug is related to the importance of angiogenesis in this tumor.

Angiogenesis plays a key role in the growth, development, and progression of GB, and it is intimately related to new vessel formation [49]. This biological process is mainly triggered by hypoxia and the increment of hypoxia-inducible factor-1 (HIF-1). Hypoxia also induces an increment of pro-angiogenic factors like vascular endothelial growth factor (VEGF), fibroblast growth factors (FGFs), and angiopoietin-1 [50,51]. Many of these pro-angiogenic factors are upregulated in GB and are related to worse prognoses.

VEGF has become a cornerstone in anti-angiogenic treatment development for GB. Higher levels of VEGF mRNA have been described in the necrotic areas of GB tumor samples, promoting new vessel creation and cell proliferation [52]. Moreover, overexpression of VEGF-R1 in low-grade astrocytomas has been related to a worse prognosis, similar to the one exhibited by high-grade gliomas [53]. Hence, VEGF expression could play a role as a prognostic marker.

Considering angiogenesis as one of the essential hallmarks of GB, anti-angiogenic therapies have been deeply explored. Among them, bevacizumab (a VEGF-A-targeting monoclonal antibody) endures as the most extensively studied anti-angiogenic therapy in GB.

In the last two decades, many trials have been carried out exploring bevacizumab in GB. FDA approval of bevacizumab for recurrent glioblastoma in 2009 was based on the results of the BRAIN phase II trial, where 167 patients who had progressed to previous temozolomide were recruited [54]. It was a non-comparative trial where patients were randomized to receive either bevacizumab 10 mg/kg alone or in combination with irinotecan 340 mg/m^2^ (or 125 mg/m^2^ in case they were receiving enzyme-inducing antiepileptic drugs) every two weeks. Objective response rates (ORR) for the bevacizumab-alone arm was 28.2% and for the combination arm 37.8%. Six-month PFS rates were 42.6% and 50.3%, respectively, and the median OS for monotherapy therapy was 9.2 months, while for the combination arm was 8.7.

Kreisl et al. explored another strategy in a phase II trial where 48 patients were initially treated with bevacizumab 10 mg/kg every two weeks and, after tumor progression, received combination therapy with irinotecan 340 mg/m^2^ or 125 mg/m^2^ every two weeks [55]. The 6-month PFS rate was 24%, and the 6-month survival rate was 57%. Interestingly, early magnetic resonance imaging (MRI) scan responses (96 h and 4 weeks after treatment initiation) were predictive of better PFS compared to stable disease (SD). Moreover, decreased cerebral edema was described in 50% of patients, and more than 50% decreased corticosteroid requirements and experienced an improvement in neurological symptoms. Only 19 patients participated in the second part of the trial, receiving a bevacizumab and irinotecan combination. None of them experienced objective responses, and 18 of them progressed after two cycles.

Brandes et al. faced bevacizumab to fotemustine in the previously mentioned phase II trial AVAREG. The 6-month OS rate was 62.1% with bevacizumab and 73.3% with fotemustine, and the median OS was 7.3 months for bevacizumab and 8.7 months for fotemustine. This study showed similar survival rates with bevacizumab and fotemustine and, therefore, the activity of bevacizumab in recurrent glioblastoma in a context of comparing bevacizumab prospectively with known active treatments for this clinical context [47].

Combinations with lomustine have also been explored in the previously mentioned BELOB trial, a controlled phase II trial [56], with three comparison groups: lomustine and bevacizumab in monotherapy or a combination of both. A total of 153 patients were included. The primary endpoint was a 9-month OS. In the lomustine group, it was 43%; (95% CI 29–57); in the bevacizumab group, 38% (95% CI 25–51, and in the combination group, 59% (43–72) with lomustine 90 mg/m^2^, 87% (39–98) with lomustine 110 mg/m^2^ and 63% (49–75) for the combined bevacizumab and lomustine group. Outcomes by MGMT status were reported, revealing a longer OS in patients with methylated MGMT promoter who received bevacizumab. The combination of bevacizumab and lomustine met the prespecified criteria to be evaluated in further phase 3 study.

Nevertheless, the most important trial carried out with this combination in the recurrent setting was not able to validate this OS benefit. In the phase 3 EORTC trial, recruited 437 patients who had progressed to chemoradiotherapy (at least 3 months after radiotherapy) and were randomized to receive lomustine 90 mg/m^2^ every 6 weeks and bevacizumab 10 mg/kg every 2 weeks versus lomustine 110 mg/m^2^ every 6 weeks in monotherapy. The median OS for bevacizumab and lomustine was 9.1 months compared to lomustine monotherapy with 8.6 months (hazard ratio 0.95, confidence interval 0.74–1.21). The combination showed improved PFS (4.2 months versus 1.5 months) and a better response rate than lomustine alone (41.5% versus 13.9%). Notwithstanding, no OS benefit was observed. Moreover, the addition of bevacizumab neither improved the time to deterioration in health-related quality of life nor neurocognitive functioning, and there was no significant difference in the time before starting glucocorticoids. The crossover to bevacizumab in the control group was 35.5%. The study concluded that the combination did not confer a survival advantage, despite it being a benefit in PFS [57].

The role of bevacizumab throughout successive lines was analyzed in the TAMIGA trial [58]. It was a phase II, randomized, placebo-controlled trial where patients with newly diagnosed GB received after surgery a first-line treatment consisting of radiotherapy plus temozolomide and bevacizumab, followed by six cycles of bevacizumab and temozolomide and finally bevacizumab until progression. After progression, randomization was performed, and patients would receive lomustine plus bevacizumab or lomustine monotherapy. New randomization occurred after the second progression when patients could be treated either with bevacizumab and chemotherapy of the investigator’s choice or chemotherapy plus placebo. The primary endpoint was OS from randomization. No survival benefit was observed from adding bevacizumab to the second and third lines, and the median OS for patients who were treated with bevacizumab was 6.4 months, lower than reported in the BRAIN trial (9.2 months) [54].

Regarding toxicity, classic adverse events associated with bevacizumab, such as hypertension, thromboembolism, and proteinuria, have been reported when used in recurrent GB patients. Among them, intracranial hemorrhage (ICH) should be outlined. Around 2–3% of patients with GB experience ICH with bevacizumab when not combined with anticoagulation [59]. Norden et al. evaluated the safety of concurrent bevacizumab and anticoagulation in a retrospective analysis, pointing out a significant increment in IHC rate when both therapies were used. Hence, the employment of bevacizumab and anticoagulation concurrently should be thoroughly evaluated [60].

Several meta-analyses have tried to unravel the role of bevacizumab in recurrent GB. Lombardi et al. evaluated the employment of bevacizumab both in first-line and recurrent GB. The analysis did not show OS improvement compared to cytotoxic treatment. Moreover, inferior outcomes were reported when bevacizumab was employed as a single agent compared to chemotherapy alone [61]. The Cochrane Collaboration performed another meta-analysis where no OS benefit was described for treatment combinations with bevacizumab for the first recurrence of patients with GBM, albeit PFS was improved with bevacizumab [62]. Likewise, Zhang et al. confirmed in their meta-analysis the absence of OS benefit for this set of patients, although an ORR positive effect was reported, and the results showed a possible benefit for PFS [63].

Several controversies remain present. Bevacizumab was approved by the FDA in 2009 but not by the European Medication Agency (EMA) for recurrent glioblastoma. The lack of approval by de EMA was because of the absence of a control arm in the pivotal trial and the uncertain validity of the outcome measurements as a surrogate of a clinical benefit. Indeed, one important issue is whether PFS can be a surrogate marker for OS. This is especially important due to the possible effect of bevacizumab on PFS that could be a reflection of the drug masking radiological disease progression. Adding to that, different trials could not demonstrate a benefit in first-line treatment.

In terms of biomarkers, until now, there are no predictive biomarkers for bevacizumab efficacy in glioblastoma or other types of tumors. The plasma value of VEFG-A and VEGFR-2, as well as a gene signature profile, are among the most studied ones.

### 4.3. Tyrosine Kinase Inhibitors (TKI)

Tyrosine kinase inhibitor (TKI) therapy in GB still remains a challenge, mainly due to the poor diffusion of these agents into tumor tissue through the blood–brain barrier (BBB) [64], as well as the presence of multiple steps and pathways involved in the oncogenic transformation of glial cells in GB biology. However, TKIs are beginning to gain special interest in recent years in the treatment of GB due to the potential for inhibition of one or more of these pathways.

#### 4.3.1. Vascular Endothelial Growth Factor (VEGF) Inhibitors

VEGFR is overexpressed on most endothelial cells in GB, making it an especially vascularized tumor. Its amplification is detected in around 6–17% of GB.

The first studies on the role of inhibition of the VEGF pathway in the treatment of recurrent GB began in the 1990s. After several phase II studies, the anti-VEGFR monoclonal antibody bevacizumab demonstrated a benefit in PFS in the treatment of GB, both in combination and as monotherapy.

Two phase II studies led to the accelerated FDA approval of bevacizumab for the treatment of GBM in 2009, as previously seen.

Regarding the role of TKIs in the inhibition of the VEGF pathway, the activity of several molecules has been extensively investigated, unfortunately without evidencing significant benefit in any of them to date.

In 2019, the previously mentioned phase II clinical trial REGOMA explored the efficacy of regorafenib, a multikinase inhibitor, compared with lomustine. The study showed an encouraging benefit with regorafenib. Median OS, its primary endpoint, was significantly improved with 7.4 months (95% CI, 5.8–12.0) vs. 5.6 months (95% CI, 4.7–7.3) in the lomustine arm [41]. A phase III study is needed to confirm these findings.

A phase III clinical trial compared cediranib (pan-VEGFR, PDGFRβ, and c-kit oral inhibitor) as monotherapy or combined with lomustine versus lomustine alone; a total of 325 patients with recurrent GB. This was a negative study that did not meet its primary endpoint of PFS with cediranib either in monotherapy or in combination [65].

A phase II study evaluated the efficacy and tolerability of the oral TKI sunitinib in monotherapy in patients with recurrent GB, stratifying by previous treatment and disease progression with bevacizumab (N = 32) or not (N = 31). Continuous daily sunitinib did not prolong PFS in both bevacizumab-naïve and bevacizumab-resistant cohorts [66]. In another phase II study testing efficacy of sunitinib in combination with irinotecan in recurrent GB, no benefit was observed in PFS [67].

Other molecules studied are vatalanib, tivozanib, sorafenib (in combination with temsirolimus and in combination with Erlotinib), and pazopanib (in monotherapy and combination with lapatinib), without favorable results [68,69,70,71,72,73].

There is currently an ongoing phase I/II clinical trial that aims to evaluate the safety and efficacy of the combination of sorafenib and everolimus in recurrent GB (NCT01434602).

#### 4.3.2. Epidermal Growth Factor Receptor (EGFR) Inhibitors

The presence of amplification, overexpression, or mutation of the EGFR pathway in GB is present in approximately 45% of cases [74] and correlates with a poor clinical prognosis, conferring chemoresistant potential. Thus, EGFR might be used as a potential prognostic biomarker [75]. Multiple EGFR-targeted TKIs have been tested in the treatment of recurrent GB.

First-generation reversible small molecule inhibitors such as erlotinib, lapatinib, and gefitinib have been tested both in monotherapy [76] and in combination with temozolomide or radiotherapy, with discouraging results in primary or recurrent GB.

Concerning the second-generation irreversible blockers, afatinib and dacomitinib have been tested in GB. Afatinib in monotherapy and in combination with temozolomide has shown limited activity, without PFS improvement, but a good safety profile in unselected GB patients [77]. On the other hand, dacomitinib is a pan-EGFR inhibitor targeting ERBB2 and ERBB4 as single-agent showed limited activity in recurrent GB [78]. Based on these results, the antitumor activity of the second-generation inhibitors is comparable to that of the first-generation.

Regarding third-generation irreversible EGFR TKIs, they are currently under investigation. Osimertinib exhibits excellent blood–brain barrier (BBB) penetration and overcomes primary resistance by blocking ERK signaling in preclinical models [79], making it a promising candidate for inhibiting EGFR in GB. A retrospective study in 15 patients with recurrent GB explored if the response to osimertinib in combination with bevacizumab could be predicted by EGFRvIII mutation in association with the EGFR gene. The results showed marginal activity in most patients, but there was a subgroup with long-lasting benefits [80]. These findings justify the continuation of the research in a clinical trial.

In 2019, the phase II clinical trial INTELLANCE 2/EORTC 1410 studied the role of depatuxizumab-mafodotin (a tumor-specific antibody-drug conjugate comprised of an antibody [ABT-806] targeting EGFR, and the toxin monomethylauristatin-F) in 260 patients with EGFR amplified GB at first recurrence after chemo-irradiation with temozolomide. Patients were randomized to either Depatux-M 1.25 mg/kg every 2 weeks intravenously monotherapy, or combined with temozolomide, or either lomustine or temozolomide as the control arm. In the long-term follow-up analysis, the median OS was 9.6 months in the combined arm (Depatux-M and Temozolomide) vs. 8.2 months in the lomustine or temozolomide arm, with an HR of 0.66 (95% CI = 0.48–0.93; *p* = 0.017). The most frequent grade 3–4-related toxicity with Depatux-M was corneal epitheliopathy, occurring in 25–30% of patients [81].

However, a phase III trial in patients with newly diagnosed GB (EGFR-amp) did not demonstrate an OS benefit for Depatux-M [82].

#### 4.3.3. Platelet-Derived Growth Factor Receptor (PDGFR) Inhibitors

The PDGFR pathway is amplified in approximately 15% of GBs [83]. Currently, multiple inhibitors are being tested in the treatment of GB.

Imatinib mesylate is a small molecule with PDGFR, KIT, and ABL inhibitory activity. However, it has not shown benefit in studies carried out for the treatment of recurrent GB either in monotherapy or in combination with hydroxyurea [84]. Subsequently, in an in vitro study together with nilotinib, it showed an increase in the migration and invasion potential of GB cells via ABL-independent stimulation of p130Cas and FAK signaling [85], an event that would explain the failure of imatinib in previous studies.

Tandutinib, an oral PDGFRβ inhibitor, showed no efficacy in trials in recurrent GB, although patients were not selected based on the presence of overexpression in the PDGFR pathway [86].

#### 4.3.4. Mesenquimal-Epithelial Transition (MET) Inhibitors

The MET gene encodes for hepatocyte growth factor. It has been implicated in the migration and invasiveness of glioma cells, and it is commonly expressed in glioblastoma [87,88].

Rilotumumab is an antibody that blocks the interaction of HGF with the c-Met receptor and has been tested in a phase II trial, with 36 patients treated with rilotumumab and bevacizumab, without benefits compared with bevacizumab, indicating that neutralizing the ligand is not effective [89].

Onartuzumab is an antibody that blocks the receptor, with no clinical benefit in recurrent disease in a phase II study combined with bevacizumab [90]

Cabozantinib, crizotinib, and INC280 are other MET inhibitors tested in clinical trials in GB.

Cabozantinib is a multitarget oral TKI with anti-MET and anti-VEGFR2 activity. In patients with recurrent GB naïve to antiangiogenic therapy, a phase II clinical trial did not show a statistical benefit; nevertheless, PFS at 6 months showed modest clinical activity with a 17.6% response rate [91].

Phase Ib GEINO-1402, whose results were published in 2022, evaluated the efficacy and tolerability of the combination therapy with temozolomide, crizotinib (a TKI with activity anti-ALK and anti-MET), and radiotherapy in 38 patients with newly diagnosed GB, showing promising results in terms of efficacy and good tolerability profile [92]. Based on these results, further investigation into the combination of ALK/MET inhibitors with chemoradiotherapy is required.

Another phase Ib clinical trial is currently underway, evaluating the efficacy and tolerability of the c-MET inhibitor capmatinib (INC280) added to bevacizumab in unresectable or recurrent GB.

#### 4.3.5. Fibroblast Growth Factor Receptor (FGFR) Inhibitors

Amplifications, fusions, and mutations of the FGFR pathway in GB are unusual and present in approximately 8% of GB cases. The most common alteration is the oncogenic chromosomal translocation of an FGFR1 or FGFR3 gene to the coding domain of TACC1 or TACC3, which leads to the FGFR kinase activation [93]. Furthermore, a preclinical study observed that GB can evade both EGFR and MET inhibition via FGFR-SPRY2 bypass signaling. Therefore, the addition of an FGFR inhibitor may increase GB response to EGFR and MET inhibition [94].

Igrafitinib is an FGFR 1–3 oral inhibitor tested in monotherapy in phase II clinical trial in recurrent GB with alterations in the FGFR pathway, showing limited efficacy but durable disease control in four cases, lasting more than 1 year in patients with tumors harboring FGFR1 or FGFR3 point mutations, or FGFR3-TACC3 fusions [95].

Erdafitinib is a potent pan-FGFR oral inhibitor, showing inhibition of cell proliferation in IDH wild-type GBM cells in studies in vivo. It has been reported that two patients with FGFR3-TACC3 rearrangements treated with erdafitinib had clinical improvement with stable disease and minor response, respectively [96].

#### 4.3.6. Other Inhibitors

Neurotrophic tyrosine kinase (NTRK) genes oncogenic fusions have been detected in different types of solid tumors in variable proportions [97]. There are two oral TKis harboring NTRK fusions, larotrectinib (a pan-TRK inhibitor) and entrectinib (with anti-pan-TRK, anti-ROS1, and anti-ALK activity), which are both FDA-approved therapies. The National Comprehensive Cancer Network (NCCN) guidelines for glioma in 2021 outline larotrectinib and entrectinib as therapies for GB with NTRK gene fusions. There are published clinical trials in patients with different solid tumors, including primary CNS tumors, with NTRK fusion [98] and entrectinib [99], that showed prolonged responses in patients with recurrent GB. These encouraging results have led to the development of second-generation NTRK inhibitors (e.g., repotrectinib, LOXO-195-BAY2731954) with lower tendencies to tumor resistance [100]

The ang-2/Tie2 pathway is overexpressed in some cases of GB in an unknown proportion [101]. It is involved in cell proliferation, tumor growth and invasion, and angiogenesis in GB. There are some studies in vitro with selective Tie2 inhibitors such as rebastinib, Bay-823, or altiratinib with promising potential.

### 4.4. Immunotherapy

Immunotherapy has revolutionized the management of advanced solid tumors. Nevertheless, the observed efficacy of immunotherapy in GB in the different published studies is lower than in other malignancies. This may be explained by the intensely immunosuppressive tumor microenvironment (TME) of GB, its low median TMB, its lack of infiltrating lymphocytes, together with our poor knowledge about the mechanisms of the immune response against this ‘immune-privileged’ disease [102].

Among these treatments would be de Immune Checkpoint Inhibitors (ICI), the treatment with oncolytic viruses (OVs), therapeutic vaccines, and adoptive cell therapy.

ICIs are monoclonal antibodies that help to restore immune response by interacting with the programmed cell death protein (PD-1) and its ligand (PDL1). They are administered intravenously.

OVs mechanism of action includes either the capacity to selectively kill tumor cells or induce specific antitumor immunity.

In the case of therapeutic cancer vaccines, several approaches of vaccination—mainly peptide, DNA, cell, and mRNA vaccines—are under evaluation to increase the immunogenicity of advanced solid tumors.

In addition, different forms of adoptive cell therapy (ACT), mainly CAR T cells, engineered TCRs, and TIL therapy, are emerging strategies of immunotherapy for advanced solid tumors.

#### 4.4.1. Immune Checkpoint Inhibitors

The ICI has been explored in patients with GB by several clinical trials, with modest results in monotherapy and combined with other treatments [103]. The results of phase II and III trials have been summarized in Table 1.

The CheckMate-548 phase III study [104] has evaluated chemoradiotherapy with TMZ plus nivolumab or placebo in 716 patients with newly diagnosed GB with methylated MGMT promoter, without significant differences in median PFS (10.6 vs. 10.3 months) and median OS (28.9 vs. 32.1 months). Another phase III trial (CheckMate-498 [105]) has compared nivolumab versus TMZ in combination with radiotherapy in 560 patients with newly diagnosed MGMT-unmethylated GB, with negative results—similar ORR and median PFS. The median OS in the group treated with TMZ was 14.9 and with nivolumab, 13.4 months.

In the CheckMate-143 phase III study [106], 369 patients with GB at first recurrence—following standard radiation and TMZ therapy—were randomized to anti-PD1 nivolumab 3 mg/kg or bevacizumab 10 mg/kg, with comparable median OS (9.8 months vs. 10.0 months) and a lower ORR in the experimental arm (7.8% vs. 23.1%).

In a cohort of 26 patients with PDL1+ GB from the basket phase I trial Keynote-028 [107], there were two partial responses (PR) to pembrolizumab (ORR 8%) lasting 8.3 and 22.8 months, with a 6-month PFS of 37.7%. Al-Harbi et al. [108] reported a major durable response to nivolumab in a pediatric patient with refractory GB and constitutional deficient mismatch repair (dMMR) due to an MSH6 homozygous mutation. These results suggest that anti-PD1 monotherapy may induce durable antitumor activity in a small subset of patients selected by immune biomarkers.

Several studies have evaluated the combination of anti-PD1 plus anti-CTLA4 antibodies. In an exploratory phase I cohort from the Checkmate-143 trial [109], 40 patients were randomized to nivolumab or nivolumab plus ipilimumab, with a similar ORR (10%) and worse tolerance in the combination group (20% of events leading to discontinuation versus 10% in the monotherapy arm). A phase I study using ipilimumab and/or nivolumab plus TMZ in newly diagnosed GB or gliosarcoma is currently ongoing (NCT02311920).

**Table 1 cancers-15-04279-t001:** Published phase II and III trials with immunotherapy in GB.

Clinical Trial	Treatment	N	Age Range	Outcomes	Relevant Toxicities
Immune checkpoint inhibitors					
Lim et al. (phase III) (CheckMate-548) [104]	TMZ + RT ± nivo	716	18–81	mPFS 10.6 m (nivo) vs. 10.3 m (pbo); mOS; mOS 31.3 m vs. 33.0 m	Grade 3/4 AEs: 52.4% (nivo) vs. 33.6% (pbo)
Omuro et al. (phase III) (CheckMate-498) [105]	Nivo + RT vs. TMZ + RT	560	18–83	mPFS 6.0 m (nivo) vs. 6.2 m (TMZ); mOS 13.4 m vs. 14.9 m; ORR 7.8% vs. 7.2%	Grade 3/4 AEs: 21.9% (nivo) vs. 25.1% (TMZ)
Reardon et al. (phase III) (CheckMate-143) [106]	Nivo vs. beva (after CT-RT)	369	22–77	mPFS 1.5 m (nivo) vs. 3.5 m (beva); mOS 9.8 m vs. 10.0 m; ORR 7.8% vs. 23.1%	Grade 3/4 AEs: 18.1% (nivo) vs. 15.2% (beva)
Nayak et al. (phase II) [110]	Pembro + beva (A) vs. pembro (B)	80	42–62 (IQR)	PFS at 6 m: 26% (A) vs. 6.7% (B); mOS 8.8 m vs. 10.3 m; ORR 20% vs. 0%	Grade 3/4 AEs: 32% (A) vs. 13% (B)
Cloughesy et al. (phase II) [111]	Pembro (neo + adj [A] vs. adj [B])	35	57.4 (mean)	mPFS 3.3 m (A) vs. 2.4 m (B); mOS 13.7 m vs. 7.5 m	Grade 3/4 AEs: 32% (A) vs. 13% (B)
Schalper et al. (phase II) [112]	(Neo)adjuvant nivo	30	NA	mPFS 4.1 m; mOS 7.3 m; higher immune cell infiltration and TCR diversity	No grade 3/4 AEs
Weathers et al. (phase I/II) [113]	Atezo + TMZ + RT	60	NA	mPFS 9.7 m; mOS 17.1 m	Grade 3/4 AEs: 55%
Reardon et al. (phase II) [114]	Nivo + varlilumab	28	NA	mOS 9.7 m; DCR 39.3% (2 PR, 9 SD)	NA
Oncolytic viruses and vaccines					
Todo et al. (phase II) [115]	G47 delta (HSV-1)	19	25–73	mOS 20.02 m; OS at 1 y: 84.2%, DCR 100% (18 SD, 1 PR lasting > 2 y)	Grade 3 AEs: 26.3% Grade 4 AEs: 10.5%
Weller et al. (phase III) (ACT IV) [116]	Rindopepimut (EGFRvIII vaccine) vs. pbo	745	51–64 (IQR)	No significant difference in OS (HR 1.01; *p* = 0.93)	Serious AEs: seizure (5% vs. 6%) and brain edema (2% vs. 3%)
Liau et al. (phase III) [117]	TMZ ± DCvax (after CT-RT)	331	19–73	mOS (ITTp) 23.1 m; OS at 1 y: 89.3%; OS at 2 y: 46.2%	Similar rate of AEs in both groups
Narita et al. (phase III) [118]	Personalized peptide vaccine (PPV) vs. BSC	88	20–74	No significant difference in OS (8.4 m vs. 8.0 m)	Grade 3/4 AEs: 39.7% vs. 36.7%

The table summarizes the clinical trials, treatment administered, number of treated patients, age range, outcomes, and toxicities.

The combination of PD1 blockade plus bevacizumab has also shown limited benefit. In a phase II trial [110], 80 bevacizumab-naïve patients with recurrent GB were randomized to pembrolizumab with or without bevacizumab. In the combination arm, there was a PFS-6 of 26%, an ORR of 20%, and a median OS of 8.8 months, compared to a PFS-6 of 6.7%, a median OS of 10.3 months and an ORR of 0% in the monotherapy group.

Atezolizumab has been explored in 16 patients with recurrent GB by a phase IA clinical trial [119], reporting 1 PR (5.3 months) and three cases of stable disease (SD). There were three patients with >16 months survival, two with IDH1 mutations, and one with a POLE-mutant tumor. This study suggested a correlation between response to atezolizumab and the levels of peripheral CD4+ T cells.

A phase II study has evaluated the combination of nivolumab with anti-CD27 agonist antibody varlilumab in 28 patients with recurrent GB, with a median OS rate of 9.7 months, 2 PR—both in patients with unmethylated MGMT—and 9 cases of stable disease (SD) [114]. Several additional trials are currently ongoing to evaluate the efficacy of ICI in combination with other agents in GB, such as atezolizumab plus ipatasertib (NCT03673787), dual anti-PD1 plus anti-Tim3 blockade (NCT03673787) and anti-PD1 plus anti-Lag3 blockade (NCT02658981).

Interestingly, some studies have suggested that the neoadjuvant administration of PD1 blockade might represent a more efficient approach [111,112].

#### 4.4.2. Oncolytic Viruses

OVs are a novel approach to immunotherapy against solid tumors. They induce immunogenic death of cancer cells, releasing tumor-associated antigens (TAA) and damage-associated molecular patterns (DAMPs) that activate innate immunity and improve the antigen cross-presentation, unleashing adaptive immune responses [120]. Different kinds of OVs (adenovirus, herpes simplex, measles virus, parvovirus, poliovirus, and zika virus) have shown preclinical efficacy against GB cells [121]. Preliminary results from early clinical trials are encouraging, with an adequate tolerance and promising activity.

G47 delta (DELYTACT)—an oncolytic variant of herpes simplex virus 1 (HSV-1)—is the only agent evaluated in GB by a phase II clinical trial so far [115]. Nineteen adult patients with residual or recurrent supratentorial GB were administered G47 delta intratumorally for up to six doses. The 1-year OS rate was 84.2%, and the median OS after treatment initiation was 20.2 months. Biopsies revealed an increase of effector -CD4+/CD8- T cells and a decrease of immunosuppressive -FoxP3+- lymphocytes.

A phase I trial [122] evaluated the retroviral vector vocimagene amiretrorepvec (Toca 511)—in combination with extended-release 5-fluorocytosine—in 56 patients with recurrent GB, reporting a durable response rate in 21.7% of cases. Another phase I study [123] showed similar results with tasadenoturev (DNX-2401)—a tumor-selective oncolytic adenovirus—in 37 patients with recurrent malignant glioma, reporting 3 cases of PFS at 3 years from treatment and a 3-year OS rate of 20%. Both Toca 511 and DNX-2401 had a satisfactory safety profile and will be further evaluated in phase II/III trials.

A live attenuated poliovirus type 1 vaccine (PVSRIPO) has shown promising results in a phase I study with 61 GB patients, with a 20% of long-term survivors [124]. It is currently being studied in combination with pembrolizumab by the phase II trial LUMINOS-101 [125]. Other promising agents are HSV-1 variant G207 [126] and H-1 parvovirus [127].

#### 4.4.3. Therapeutic Vaccines

Three vaccination agents have reached phase III clinical trials in GB so far: Rindopepimut, DCvax, and PPV.

Rindopepimut is a specific peptide vaccine targeting an EGFR deletion (EGFRvIII), which occurs in around 20–30% of GB and is associated with poor long-term survival [128]. Several phase II clinical trials [129] evaluating rindopepimut in patients with newly diagnosed EGFRvIII-expressing GB showed robust immune responses and promising clinical data. Unfortunately, a randomized phase III trial (ACT IV) [116] with temozolomide (TMZ) plus rindopepimut or placebo was terminated for futility after a preplanned interim analysis, with no significant difference in overall survival (HR 1.01).

DCvax is a personalized peptide vaccine that uses autologous tumor lysate-loaded dendritic cells (DCs). In the phase III clinical trial [117], 331 patients with GB were randomized, after surgery and chemoradiotherapy, to receive TMZ plus DCvax or TMZ, with a median OS of 23.1 months from surgery for the intention-to-treat (ITT) population, and 34.7 months for patients with methylated MGMT. Interestingly, around 30% of the ITT population had a particularly extended mOS (40.5 months) not explained by known prognostic factors. Overall adverse events in the DCvax group were similar to the control group.

Personalized peptide vaccination (PPV) showed promising clinical activity in a phase I trial with 15 recurrent TMZ-refractory GB [130], though a randomized phase III trial comparing PPV treatment versus best supportive had unfavorable results (median OS 8.4 months versus 8 months) [118]. The identification of neoantigens by whole exome DNA and RNA sequencing offers a more promising approach for personalized vaccination, with positive results in early studies (vaccines APVAC1/2 [130] and NeoVax [131].

Vaccines against insulin-like growth factor (IGF) type 1 receptor, surviving peptides, and IDH1 peptides are under evaluation for patients with GB, with encouraging results in pre-clinical models or early clinical trials [103].

#### 4.4.4. Adoptive Cell Therapy

In CAR T therapy—the most studied strategy in GB so far—T cells are modified in vitro to incorporate a stable high-affinity single-chain fragment variable known as chimeric antigen receptor (CAR), which is specific against a target protein. This cell product can be equipped with costimulatory receptors and other immune-enhancing molecules to overcome immunosuppressive TME [103]. Several kinds of CAR T therapy have been evaluated in refractory GB by preclinical studies and early clinical trials, with modest results, including CAR T products against interleukin-13 receptor subunit alpha-2 (IL13 Rα2), EGFRvIII, B7H3/CD276, and Her2 [117,132,133].

## 5. NovoTTF-100A

NovoTTF-100A is a portable device that, through transducer arrays, delivers low-intensity, intermediate-frequency electric fields. The device is applied to a shaved scalp and connected to a portable battery. Continuous treatment is recommended.

At a molecular level, these alternating electric fields lead to impaired cytokinesis and asymmetric chromosome segregation resulting in mitotic cell death.

A phase III trial compared NovoTTF versus active chemotherapy in patients with recurrent glioblastoma. Median survival was 6.6 months vs. 6 months, the 1-year survival rate was 20% in both arms, and the PFS rate at 6 months was 21.4 vs. 15.1 (*p* = 0.13) with TTF vs. active arm. Adverse events were mild or moderate, mainly related to skin rash. Therefore, the efficacy of the device was comparable to chemotherapy treatments used in recurrent glioblastoma [134].

## 6. Drug Delivery Strategies

As described above, one of the main reasons for the poor results from the use of systemic therapy in GB is their low diffusion through the BBB. Currently, studies are being carried out to develop strategies to increase the permeability of the BBB and the use of drugs locally or intranasally, as well as the use of nanotechnology with the purpose of increasing drug concentration in the tumor and its environment [135].

## 7. Conclusions

Glioblastoma is one of the most aggressive tumors. In recent years several treatment strategies have been studied after the progression of initial treatment, including TKI, immunotherapy, vaccines, or cell therapy, some with promising results.

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
