# Peer review of "Recurrent Glioblastoma: A Review of the Treatment Options"

_cancers, 2023, doi:10.3390/cancers15174279_

Round 1

Reviewer 1 Report

Vaz-Salgado and co-authors have reviewed the recent clinical trials for the treatment of relapsed glioblastoma. Modifications are required before publication.

 Minor modifications

 1)   Revise the text to correct typographical and structural errors in certain sentences.

2)   Define abbreviations, such as TKI, OS, PFS, GB the first time they appear.

3)   Line 33: The authors wrote "surg". Did they want to write "surgery"?

4)   Lines 83 – 85: The authors are invited to add by how many months the OS was improved according to reference #12.

5)   Line 89: What neurological symptoms were improved? Have these improvements been quantified?

6)   Lines 104-105: I do not understand this sentence “In a phase II study (14) of 90 patients the OS was 17 months, OS-1 year 66.7%, OS-2 104 years 32.6% and OS-3 years 22.2%[14]”.

7)   Lines 106-108: How these results compare to those obtained when the patients were not re-irradiated. By how many months was overall survival at different time points improved?

8)   Line 112: Define “G3 toxicity”.

9)   Line 118: Define “RPA class”.

10)    Line 120: What do the authors mean by “good PS”?

11)    Line 145: It would be more appropriate to place the reference #22 just after "... with other approaches" (line 143).

12)    References # 22 and 23 are associated to the same manuscript. Is it possible that the manuscript associated with reference #23 is #24 instead?

13)    Line 151: How many months has the PFS been extended?

14)    Line 169: Replace “Lomutine”, by “Lomustine”.

15)    Line 378: Replace "Tirosine Kinasa inhibitors” by “Tyrosine kinase inhibitors”.

16)    Line 471: I don’t understand the meaning of “D” in “D. MET inhibitors”.

17)    Line 513:  I don’t understand the meaning of “F” in “F. Other inhibitors.”

18)    A title should be added to the table.

 Revise the text to correct typographical and structural errors in certain sentences.

Author Response

Attached you will find the document with the reply.

Vaz-Salgado and co-authors have reviewed the recent clinical trials for the treatment of relapsed glioblastoma. Modifications are required before publication.

 Minor modifications

 1)   Revise the text to correct typographical and structural errors in certain sentences.

            Done

2)   Define abbreviations, such as TKI, OS, PFS, GB the first time they appear.

The corrections have been made           

3)   Line 33: The authors wrote "surg". Did they want to write "surgery"?

            The corrections have been made

4)   Lines 83 – 85: The authors are invited to add by how many months the OS was improved according to reference #12.

            These improvements were not quantified

5)   Line 89: What neurological symptoms were improved? Have these improvements been quantified?

            Done

6)   Lines 104-105: I do not understand this sentence “In a phase II study (14) of 90 patients the OS was 17 months, OS-1 year 66.7%, OS-2 104 years 32.6% and OS-3 years 22.2%[14]”.

            The corrections have been made

7)   Lines 106-108: How these results compare to those obtained when the patients were not re-irradiated. By how many months was overall survival at different time points improved?

            The corrections have been made

8)   Line 112: Define “G3 toxicity”.

            The corrections have been made

9)   Line 118: Define “RPA class”.

            The corrections have been made

10)    Line 120: What do the authors mean by “good PS”?

            The corrections have been made

11)    Line 145: It would be more appropriate to place the reference #22 just after "... with other approaches" (line 143).

            The corrections have been made

12)    References # 22 and 23 are associated to the same manuscript. Is it possible that the manuscript associated with reference #23 is #24 instead?

            The corrections have been made

13)    Line 151: How many months has the PFS been extended?

            The corrections have been made

14)    Line 169: Replace “Lomutine”, by “Lomustine”.

            The corrections have been made

15)    Line 378: Replace "Tirosine Kinasa inhibitors” by “Tyrosine kinase inhibitors”.

            The corrections have been made

16)    Line 471: I don’t understand the meaning of “D” in “D. MET inhibitors”.

            The corrections have been made

17)    Line 513:  I don’t understand the meaning of “F” in “F. Other inhibitors.”

            The corrections have been made

18)    A title should be added to the table.

            The corrections have been made

Reviewer 2 Report

Dear Authors,

After carefully reviewing your article entitled "Recurrent glioblastoma, a review of the treatment options" I have found some points needed to be addressed. 

Firstly, the use of a "simple summary" along with abstract should be clarified, as well as the need to ensure linguistic accuracy in the abstract section.

The introduction should provide more comprehensive background information on glioblastoma and the available therapeutic options, including a discussion of the impact on patients' quality of life and the challenges associated with managing the disease.

Merge short paragraphs into full-length statements and restructure the information on therapeutic strategies more meticulously, incorporating radiotherapy, surgery, systemic treatments, and immunotherapy. The lack of assigned table numbers and captions should also be rectified.

The section on surgery could benefit from the inclusion of specific studies, discussing their findings and conclusions regarding the impact of surgery on survival in recurrent glioblastoma. Furthermore, complications, quality of life outcomes, emerging surgical techniques, and future directions should be elaborated upon.

Regarding radiotherapy, more details are needed on prospective data for reirradiation and the challenges associated with conducting trials in this context. The section should discuss adjuvant radiotherapy after resection of recurrences, different fractionation schemes, the role of brachytherapy, and systemic therapy combinations. Future research directions and some additional ongoing clinical trials should also be addressed.

In the section on systemic treatment, a comparative analysis of lomustine and fotemustine should be included, along with safety profiles and information on combination therapies and emerging treatment options. The discussion on bevacizumab should cover its mechanism of action, biomarkers, adverse events, controversies, and ongoing research. The section on tyrosine kinase inhibitors should discuss BBB penetration, combination therapies, resistance mechanisms, personalized medicine approaches, and future directions.

In the sections on immunotherapy, oncolytic viruses, therapeutic vaccines, adoptive cell therapy, NovoTTF-100A, and drug delivery strategies, additions are needed to provide more in-depth explanations of mechanisms of action, clinical implications, ongoing developments, and future prospects for each treatment modality.

Overall, incorporating the suggested revisions will result in a more comprehensive and informative review article on the treatment options for recurrent glioblastoma.

Thanks.

Reviewer.

Little linguistic improvement is needed. 

Author Response

Attached you will find the document with the reply.

Reviewer 2

Dear Authors,

After carefully reviewing your article entitled "Recurrent glioblastoma, a review of the treatment options" I have found some points needed to be addressed. 

Firstly, the use of a "simple summary" along with abstract should be clarified, as well as the need to ensure linguistic accuracy in the abstract section.

The “simple summary” is required by the journal. Linguistic accuracy has been reviewed

The introduction should provide more comprehensive background information on glioblastoma and the available therapeutic options, including a discussion of the impact on patients' quality of life and the challenges associated with managing the disease.

We have done a more comprehensive introduction

Merge short paragraphs into full-length statements and restructure the information on therapeutic strategies more meticulously, incorporating radiotherapy, surgery, systemic treatments, and immunotherapy. The lack of assigned table numbers and captions should also be rectified.

These aspects have been reviewed

The section on surgery could benefit from the inclusion of specific studies, discussing their findings and conclusions regarding the impact of surgery on survival in recurrent glioblastoma. Furthermore, complications, quality of life outcomes, emerging surgical techniques, and future directions should be elaborated upon.

These aspects have been reviewed

Regarding radiotherapy, more details are needed on prospective data for reirradiation and the challenges associated with conducting trials in this context.

Unfortunately there is currently no further evidence about reirradiation of glioblastoma. Most studies are retrospectiv

The section should discuss adjuvant radiotherapy after resection of recurrences,

We have included two retrospective study on the benefit of radiotherapy after surgery. The ongoing Gliocave trial was already included in this paper.

  1. Re-irradiation after groos total resection of recurrent glioblastoma. Straube et al. Sthalenther Onkol 2017. 193:897-909
  2. Survival gain with re-Op/RT for recurred high-grade gliomas depends upon risk groups. Chun et al 2018. Radiotherapy and Oncology 128: 254-259.

different fractionation schemes,

We had already included the most important fractionation schemes

 the role of brachytherapy,

We had already included the scientific evidence on this topic

and systemic therapy combinations.

We had already included the combination of bevacizumab and reirradiation.

We have included a retrospective study about reirradiation and TMZ combination

  • Reirradiation for recurrent glioblastoma multiforme: a critical comparison of different concepts. Sthalenther Onkol 2020: 196 (457-464)

 There is little strong evidence about the combination of others systemic treatment and reirradiation.

Future research directions and some additional ongoing clinical trials should also be addressed.

The most current evidence and the few ongoing trials on the subject had already been included in the review.

In the section on systemic treatment, a comparative analysis of lomustine and fotemustine should be included, along with safety profiles and information on combination therapies and emerging treatment options.

A comparative analysis of lomustine and fotemustine has been done, as well as safety profiles, combination and emerging options

The discussion on bevacizumab should cover its mechanism of action, biomarkers, adverse events, controversies, and ongoing research.

These aspects have been introduced in the article

The section on tyrosine kinase inhibitors should discuss BBB penetration, combination therapies, resistance mechanisms, personalized medicine approaches, and future directions.

These data have been reviewed

In the sections on immunotherapy, oncolytic viruses, therapeutic vaccines, adoptive cell therapy, NovoTTF-100A, and drug delivery strategies, additions are needed to provide more in-depth explanations of mechanisms of action, clinical implications, ongoing developments, and future prospects for each treatment modality.

These suggestions have been introduced in the text

Overall, incorporating the suggested revisions will result in a more comprehensive and informative review article on the treatment options for recurrent glioblastoma.

Thanks.

Reviewer.

Reviewer 3 Report

The paper “Recurrent glioblastoma, a review of the treatment options  by Vaz-Salgado” is quite interesting and provide a broad overview especially about systemic treatments including immunotherapy with oncolytic viruses and therapeuthic vaccines. On the contrary, the aspect related to radiotherapy treatments have not been described with a similar level of detail. Stereotactic radiosurgery treatments are only hinted at to justify the diminished role of brachytherapy at present, which is increasingly being replaced by stereotactic radiosurgery. While SRS is commonly used in the definitive and adjuvant settings for other CNS malignancies, its role in the preoperative setting has become a topic of great interest due to the potential for reduced treatment volumes due to the treatment of an intact tumor, and a lower risk of nodular leptomeningeal disease and radiation necrosis. Preclinical data suggest that preoperative SRS may further enhance immune responses against GBM, which can be exploited to improve OS. Further studies are needed to better define the role of preoperative SRS in this context. Further preclinical and early clinical studies are underway to further evaluate this therapy.such as the NeoGlioma study (NCT05030298), which is a phase 1/2A clinical trial to evaluate the role of SRS in this setting. phase 1/2A clinical trial evaluating the role of preoperative SRS in high-grade glioma. While early reports of SRS in the adjuvant setting for glioblastoma were disappointing, its role in the preoperative setting and its impact on the anti-tumor adaptive immune response is largely unknown (Leherer et al Preoperative Stereotactic Radiosurgery for Glioblastoma. Preoperative Stereotactic Radiosurgery for Glioblastoma. Biology 2022, 11, 194. https:// doi.org/10.3390/biology11020194.

In addition, fractionated stereotactic re-irradiation produces relative clinical benefit and safety Biology 2022, 11, 194. https:doi.org/10.3390/biology11020194)  for patients with recurrent glioblastoma. (Luo et al. Fractionated stereotactic re-irradiation for recurrent glioblastoma: A systematic review and meta-analysis Clinical Neurology and Neurosurgery 229 (2023) 107728 Available)

Furthermore Proton therapy has not taken into account as possible radiotherapy treatment. For review see:

1.      Proton radiotherapy for glioma and glioblastoma  Chin Clin Oncol 2022;11(6):46 | https://dx.doi.org/10.21037/cco-22-92 

2.      Proton Therapy and Gliomas: A Systematic Review by Chambrelant et al Radiation 2021, 1, 218–233. https://doi.org/10.3390/radiation1030019

Finally, Ferroptosis is a newly found non apoptotic regulatory cell death process that plays a vital role in a variety of brain diseases, including cerebral hemorrhage, neurodegenerative diseases, and primary or metastatic brain tumors. Recent studies have shown that targeting ferroptosis can be an effective strategy to overcome resistance to tumor therapy and immune escape mechanisms. This suggests that combining ferroptosis-based therapies with other treatments may be an effective strategy to improve the treatment of GBM.

1.      Emerging role of ferroptosis in glioblastoma: Therapeutic opportunities and challenges Zhuo et al. Therapeutic opportunities and challenges. Front. Mol. Biosci. 9:974156. doi: 10.3389/fmolb.2022.974156

2.     Pyroptosis, ferroptosis, and autophagy cross‑talk in glioblastoma opens up new avenues for glioblastoma treatment Sicheng Wan. Cell Communication and Signaling (2023) 21:115

Major revisions

Authors are asked to improuve the text by discussing the three bullet points suggested

English is fine

Author Response

On the contrary, the aspect related to radiotherapy treatments have not been described with a similar level of detail. Stereotactic radiosurgery treatments are only hinted at to justify the diminished role of brachytherapy at present, which is increasingly being replaced by stereotactic radiosurgery. While SRS is commonly used in the definitive and adjuvant settings for other CNS malignancies, its role in the preoperative setting has become a topic of great interest due to the potential for reduced treatment volumes due to the treatment of an intact tumor, and a lower risk of nodular leptomeningeal disease and radiation necrosis. Preclinical data suggest that preoperative SRS may further enhance immune responses against GBM, which can be exploited to improve OS. Further studies are needed to better define the role of preoperative SRS in this context. Further preclinical and early clinical studies are underway to further evaluate this therapy.such as the NeoGlioma study (NCT05030298), which is a phase 1/2A clinical trial to evaluate the role of SRS in this setting. phase 1/2A clinical trial evaluating the role of preoperative SRS in high-grade glioma. While early reports of SRS in the adjuvant setting for glioblastoma were disappointing, its role in the preoperative setting and its impact on the anti-tumor adaptive immune response is largely unknown (Leherer et al Preoperative Stereotactic Radiosurgery for Glioblastoma. Preoperative Stereotactic Radiosurgery for Glioblastoma. Biology 2022, 11, 194. https:// doi.org/10.3390/biology11020194.

The role of preoperative SRS has been explored in the malignant glioma de novo, but it has not been studied for the recurrent scenario, where we have to consider the previous doses of radiation therapy received by surrounding organs at risk. Both references mentioned (Neo Glioma Study NTC5030298 and Lehrer et al review, are focus on the novo diagnostic of glioblastoma.

In addition, fractionated stereotactic re-irradiation produces relative clinical benefit and safety Biology 2022, 11, 194. https:doi.org/10.3390/biology11020194)  for patients with recurrent glioblastoma. (Luo et al. Fractionated stereotactic re-irradiation for recurrent glioblastoma: A systematic review and meta-analysis Clinical Neurology and Neurosurgery 229 (2023) 107728 Available)

We have included the fractionated stereotactic schedule of 30 Gy in 5 fractions, now we have included this reference.

Furthermore Proton therapy has not taken into account as possible radiotherapy treatment. For review see:

  1. Proton radiotherapy for glioma and glioblastoma  Chin Clin Oncol 2022;11(6):46 | https://dx.doi.org/10.21037/cco-22-92
  2. Proton Therapy and Gliomas: A Systematic Review by Chambrelant et al Radiation 2021, 1, 218–233. https://doi.org/10.3390/radiation1030019

 For proton therapy, the review of Chamberlant et al  is algo for the novo scenario, the Goff et al review is mainly in the novo scenario, but they mentioned the largest series of recurrent GBM treated with protontherapy, so we have included inour review, adding a study about quality of life in this same clinical scenario

Saeed AM, Khairnar R, Sharma AM et al. Clinical Outcomes in Patients with Recurrent Glioblastoma Treated with Proton Beam Therapy Reirradiation: Analysis of the Multi-Institutional Proton Collaborative Group Registry. Adv Radiat Oncol. 2020 Apr 22;5(5):978-983. doi: 10.1016/j.adro.2020.03.022. PMID: 33083661

 Scartoni D, Amelio D, Palumbo P, Giacomelli I, Amichetti M. Proton therapy re-irradiation preserves health-related quality of life in large recurrent glioblastoma. J Cancer Res Clin Oncol. 2020 Jun;146(6):1615-1622. doi: 10.1007/s00432-020-03187-w. Epub 2020 Mar 21. PMID: 32200460.

Major revisions

Authors are asked to improuve the text by discussing the three bullet points suggested

Round 2

Reviewer 2 Report

Dear Authors,

It's quite commendable that you have improved your manuscript. I have 2 suggestions for further refinement. 

1. Table title should be incorporated above table and and write something about table in a caption below table i.e. why it's there for and what information does it provide. 

2. Please write functional mechanism of device NovoTTF-100A in brief. 

Your article looks quite improved and its ready to get published after this improvement.

All the best!!

Best,

The Reviewer

Author Response

Both suggestions have been incorporated in the text

Reviewer 3 Report

The authors put a lot of effort into incorporating the comments of the referees. Although the work does not reach a high scientific level, it nevertheless makes a useful contribution to the scientific community. The work in this final version can be accepted for publication in Cancers

English is fine

Author Response

The suggestions have been considered

Thank you
